# Fast and Robust Least Squares Estimation in Corrupted Linear Models

**Brian McWilliams**[*]    **Gabriel Krummenacher**[*]    **Mario Lucic**    **Joachim M. Buhmann**
Department of Computer Science
ETH Zürich, Switzerland
{mcbrian,gabriel.krummenacher,lucic,jbuhmann}@inf.ethz.ch

## Abstract

*Subsampling methods* have been recently proposed to speed up least squares estimation in large scale settings. However, these algorithms are typically not robust to outliers or corruptions in the observed covariates.

The concept of *influence* that was developed for regression diagnostics can be used to detect such corrupted observations as shown in this paper. This property of influence – for which we also develop a randomized approximation – motivates our proposed subsampling algorithm for large scale corrupted linear regression which limits the influence of data points since highly influential points contribute most to the residual error. Under a general model of corrupted observations, we show theoretically and empirically on a variety of simulated and real datasets that our algorithm improves over the current state-of-the-art approximation schemes for ordinary least squares.

## 1   Introduction

To improve scalability of the widely used ordinary least squares algorithm, a number of randomized approximation algorithms have recently been proposed. These methods, based on subsampling the dataset, reduce the computational time from $O\left(np^2\right)$ to $o(np^2)$[1] [14]. Most of these algorithms are concerned with the classical fixed design setting or the case where the data is assumed to be sampled i.i.d. typically from a sub-Gaussian distribution [7]. This is known to be an unrealistic modelling assumption since real-world data are rarely well-behaved in the sense of the underlying distributions.

We relax this limiting assumption by considering the setting where with some probability, the observed covariates are corrupted with additive noise. This scenario corresponds to a generalised version of the classical problem of "errors-in-variables" in regression analysis which has recently been considered in the context of sparse estimation [12]. This corrupted observation model poses a more realistic model of real data which may be subject to many different sources of measurement noise or heterogeneity in the dataset.

A key consideration for sampling is to ensure that the points used for estimation are typical of the full dataset. Typicality requires the sampling distribution to be robust against outliers and corrupted points. In the i.i.d. sub-Gaussian setting, outliers are rare and can often easily be identified by examining the *statistical leverage* scores of the datapoints.

Crucially, in the corrupted observation setting described in §2, the concept of an outlying point concerns the relationship between the observed predictors and the response. Now, leverage alone cannot detect the presence of corruptions. Consequently, without using additional knowledge about

---

[*]Authors contributed equally.
[1]Informally: $f(n) = o(g(n))$ means $f(n)$ grows more slowly than $g(n)$.

the corrupted points, the OLS estimator (and its subsampled approximations) are biased. This also rules out stochastic gradient descent (SGD) – which is often used for large scale regression – since convex cost functions and regularizers which are typically used for noisy data are not robust with respect to measurement corruptions.

This setting motivates our use of *influence* – the effective impact of an individual datapoint exerts on the overall estimate – in order to detect and therefore avoid sampling corrupted points. We propose an algorithm which is robust to corrupted observations and exhibits reduced bias compared with other subsampling estimators.

**Outline and Contributions.**    In §2 we introduce our corrupted observation model before reviewing the basic concepts of statistical leverage and influence in §3. In §4 we briefly review two subsampling approaches to approximating least squares based on structured random projections and leverage weighted importance sampling. Based on these ideas we present influence weighted subsampling (`IWS-LS`), a novel randomized least squares algorithm based on subsampling points with small influence in §5.

In §6 we analyse `IWS-LS` in the general setting where the observed predictors can be corrupted with additive sub-Gaussian noise. Comparing the `IWS-LS` estimate with that of OLS and other randomized least squares approaches we show a reduction in both bias and variance. It is important to note that the simultaneous reduction in bias and variance is relative to OLS and randomized approximations which are only unbiased in the non-corrupted setting. Our results rely on novel finite sample characteristics of leverage and influence which we defer to §SI.3. Additionally, in §SI.4 we prove an estimation error bound for `IWS-LS` in the standard sub-Gaussian model.

Computing influence exactly is not practical in large-scale applications and so we propose two randomized approximation algorithms based on the randomized leverage approximation of [8]. Both of these algorithms run in $o(np^2)$ time which improve scalability in large problems. Finally, in §7 we present extensive experimental evaluation which compares the performance of our algorithms against several randomized least squares methods on a variety of simulated and real datasets.

## 2    Statistical model

In this work we consider a variant of the standard linear model

$$\mathbf{y} = \mathbf{X}\boldsymbol{\beta} + \epsilon, \tag{1}$$

where $\epsilon \in \mathbb{R}^n$ is a noise term independent of $\mathbf{X} \in \mathbb{R}^{n \times p}$. However, rather than directly observing $\mathbf{X}$ we instead observe $\mathbf{Z}$ where

$$\mathbf{Z} = \mathbf{X} + U\mathbf{W}. \tag{2}$$

$U = \text{diag}(u_1, \ldots, u_n)$ and $u_i$ is a Bernoulli random variable with probability $\pi$ of being 1. $\mathbf{W} \in \mathbb{R}^{n \times p}$ is a matrix of measurement corruptions. The rows of $\mathbf{Z}$ therefore are corrupted with probability $\pi$ and not corrupted with probability $(1 - \pi)$.

**Definition 1** (Sub-gaussian matrix)**.** *A zero-mean matrix* $\mathbf{X}$ *is called sub-Gaussian with parameter* $(\frac{1}{n}\sigma_x^2, \frac{1}{n}\Sigma_x)$ *if (a) Each row* $\mathbf{x}_i^\top \in \mathbb{R}^p$ *is sampled independently and has* $\mathbb{E}[\mathbf{x}_i \mathbf{x}_i^\top] = \frac{1}{n}\Sigma_x$. *(b) For any unit vector* $\mathbf{v} \in \mathbb{R}^p$, $\mathbf{v}^\top \mathbf{x}_i$ *is a sub-Gaussian random variable with parameter at most* $\frac{1}{\sqrt{p}}\sigma_x$.

We consider the specific instance of the linear corrupted observation model in Eqs. (1), (2) where

- $\mathbf{X}, \mathbf{W} \in \mathbb{R}^{n \times p}$ are sub-Gaussian with parameters $(\frac{1}{n}\sigma_x^2, \frac{1}{n}\Sigma_x)$ and $(\frac{1}{n}\sigma_w^2, \frac{1}{n}\Sigma_w)$ respectively,
- $\epsilon \in \mathbb{R}^n$ is sub-Gaussian with parameters $(\frac{1}{n}\sigma_\epsilon^2, \frac{1}{n}\sigma_\epsilon^2 \mathbf{I}_n)$,

and all are independent of each other.

The key challenge is that even when $\pi$ and the magnitude of the corruptions, $\sigma_w$ are relatively small, the standard linear regression estimate is biased and can perform poorly (see §6). Sampling methods which are not sensitive to corruptions in the observations can perform even worse if they somehow subsample a proportion $rn > \pi n$ of corrupted points. Furthermore, the corruptions may not be large enough to be detected via leverage based techniques alone.

The model described in this section generalises the "errors-in-variables" model from classical least squares modelling. Recently, similar models have been studied in the high dimensional ($p \gg n$)

setting in [4–6, 12] in the context of robust sparse estimation. The "low-dimensional" $(n > p)$ setting is investigated in [4], but the "big data" setting $(n \gg p)$ has not been considered so far.[2]

In the high-dimensional problem, knowledge of the corruption covariance, $\Sigma_w$ [12], or the data covariance $\Sigma_x$ [5], is required to obtain a consistent estimate. This assumption may be unrealistic in many settings. We aim to reduce the bias in our estimates *without* requiring knowledge of the true covariance of the data or the corruptions, and instead sub-sample only non-corrupted points.

## 3  Diagnostics for linear regression

In practice, the sub-Gaussian linear model assumption is often violated either by heterogeneous noise or by a corruption model as in §2. In such scenarios, fitting a least squares model to the full dataset is unwise since the outlying or corrupted points can have a large adverse effect on the model fit. *Regression diagnostics* have been developed in the statistics literature to detect such points (see e.g. [2] for a comprehensive overview). Recently, [14] proposed subsampling points for least squares based on their leverage scores. Other recent works suggest related influence measures that identify subspace [16] and multi-view [15] clusters in high dimensional data.

### 3.1  Statistical leverage

For the standard linear model in Eq. (1), the well known least squares solution is

$$\widehat{\boldsymbol{\beta}} = \arg \min_{\boldsymbol{\beta}} \|\mathbf{y} - \mathbf{X}\boldsymbol{\beta}\|^2 = \left(\mathbf{X}^\top \mathbf{X}\right)^{-1} \mathbf{X}^\top \mathbf{y}. \tag{3}$$

The projection matrix $\mathbf{I} - \mathbf{L}$ with $\mathbf{L} := \mathbf{X}(\mathbf{X}^\top \mathbf{X})^{-1}\mathbf{X}^\top$ specifies the subspace in which the residual lies. The diagonal elements of the "hat matrix" $\mathbf{L}$, $l_i := L_{ii}$, $i = 1, \ldots, n$ are the *statistical leverage* scores of the $i^{th}$ sample. Leverage scores quantify to what extent a particular sample is an outlier with respect to the distribution of $\mathbf{X}$.

An equivalent definition from [14] which will be useful later concerns any matrix $\mathbf{U} \in \mathbb{R}^{n \times p}$ which spans the column space of $\mathbf{X}$ (for example, the matrix whose columns are the left singular vectors of $\mathbf{X}$). The statistical leverage scores of the rows of $\mathbf{X}$ are the squared row norms of $\mathbf{U}$, i.e. $l_i = \|\mathbf{U}_i\|^2$.

Although the use of leverage can be motivated from the least squares solution in Eq. (3), the leverage scores do not take into account the relationship between the predictor variables and the response variable $\mathbf{y}$. Therefore, low-leverage points may have a weak predictive relationship with the response and vice-versa. In other words, it is possible for such points to be outliers with respect to the conditional distribution $P(\mathbf{y}|\mathbf{X})$ but not the marginal distribution on $\mathbf{X}$.

### 3.2  Influence

A concept that captures the predictive relationship between covariates and response is *influence*. Influential points are those that might not be outliers in the geometric sense, but instead adversely affect the estimated coefficients.

One way to assess the influence of a point is to compute the change in the learned model when the point is removed from the estimation step. [2]. We can compute a leave-one-out least squares estimator by straightforward application of the Sherman-Morrison-Woodbury formula (see Prop. 3 in §SI.3):

$$\widehat{\boldsymbol{\beta}}_{-i} = \left(\mathbf{X}^\top \mathbf{X} - \mathbf{x}_i^\top \mathbf{x}_i\right)^{-1} \left(\mathbf{X}^\top \mathbf{y} - \mathbf{x}_i^\top y_i\right) = \widehat{\boldsymbol{\beta}} - \frac{\boldsymbol{\Sigma}^{-1} \mathbf{x}_i^\top e_i}{1 - l_i}$$

where $e_i = y_i - \mathbf{x}_i \widehat{\boldsymbol{\beta}}_{\mathrm{OLS}}$. Defining the influence[3], $d_i$ as the change in expected mean squared error we have

$$d_i = \left(\widehat{\boldsymbol{\beta}} - \widehat{\boldsymbol{\beta}}_{-i}\right)^\top \mathbf{X}^\top \mathbf{X} \left(\widehat{\boldsymbol{\beta}} - \widehat{\boldsymbol{\beta}}_{-i}\right) = \frac{e_i^2 l_i}{(1 - l_i)^2}.$$

Points with large values of $d_i$ are those which, if added to the model, have the largest adverse effect on the resulting estimate. Since influence only depends on the OLS residual error and the leverage scores, it can be seen that the influence of every point can be computed at the cost of a least squares fit. In the next section we will see how to approximate both quantities using random projections.

## 4   Fast randomized least squares algorithms

We briefly review two randomized approaches to least squares approximation: the importance weighted subsampling approach of [9] and the dimensionality reduction approach [14]. The former proposes an importance sampling probability distribution according to which, a small number of rows of $\mathbf{X}$ and $\mathbf{y}$ are drawn and used to compute the regression coefficients. If the sampling probabilities are proportional to the statistical leverages, the resulting estimator is close to the optimal estimator [9]. We refer to this as `LEV-LS`.

The dimensionality reduction approach can be viewed as a random projection step followed by a uniform subsampling. The class of Johnson-Lindenstrauss projections – e.g. the SRHT – has been shown to approximately uniformize leverage scores in the projected space. Uniformly subsampling the rows of the projected matrix proves to be equivalent to leverage weighted sampling on the original dataset [14]. We refer to this as `SRHT-LS`. It is analysed in the statistical setting by [7] who also propose `ULURU`, a two step fitting procedure which aims to correct for the subsampling bias and consequently converges to the OLS estimate at a rate independent of the number of subsamples [7].

**Subsampled Randomized Hadamard Transform (SRHT)**   The SHRT consists of a preconditioning step after which $n_{subs}$ rows of the new matrix are subsampled uniformly at random in the following way $\sqrt{\frac{n}{n_{subs}}}\mathbf{SHD} \cdot \mathbf{X} = \mathbf{\Pi X}$ with the definitions [3]:
- $\mathbf{S}$ is a subsampling matrix.
- $\mathbf{D}$ is a diagonal matrix whose entries are drawn independently from $\{-1, 1\}$.
- $\mathbf{H} \in \mathbb{R}^{n \times n}$ is a normalized Walsh-Hadamard matrix[4] which is defined recursively as

$$\mathbf{H}_n = \left[ \begin{array}{cc} \mathbf{H}_{n/2} & \mathbf{H}_{n/2} \\ \mathbf{H}_{n/2} & -\mathbf{H}_{n/2} \end{array} \right], \ \ \mathbf{H}_2 = \left[ \begin{array}{cc} +1 & +1 \\ +1 & -1 \end{array} \right].$$

We set $\mathbf{H} = \frac{1}{\sqrt{n}}\mathbf{H}_n$ so it has orthonormal columns.

As a result, the rows of the transformed matrix $\mathbf{\Pi X}$ have approximately uniform leverage scores. (see [17] for detailed analysis of the SRHT). Due to the recursive nature of $\mathbf{H}$, the cost of applying the SRHT is $O\left(pn \log n_{subs}\right)$ operations, where $n_{subs}$ is the number of rows sampled from $\mathbf{X}$ [1].

The `SRHT-LS` algorithm solves $\widehat{\boldsymbol{\beta}}_{SRHT} = \arg\min_{\boldsymbol{\beta}} \|\mathbf{\Pi y} - \mathbf{\Pi X}\boldsymbol{\beta}\|^2$ which for an appropriate subsampling ratio, $r = \Omega(\frac{p^2}{\rho^2})$ results in a residual error, $\tilde{\mathbf{e}}$ which satisfies

$$\|\tilde{\mathbf{e}}\| \leq (1 + \rho)\|\mathbf{e}\| \tag{4}$$

where $\mathbf{e} = \mathbf{y} - \mathbf{X}\widehat{\boldsymbol{\beta}}_{\text{OLS}}$ is the vector of OLS residual errors [14].

**Randomized leverage computation**   Recently, a method based on random projections has been proposed to approximate the leverage scores based on first reducing the dimensionality of the data using the SRHT followed by computing the leverage scores using this low-dimensional approximation [8–10, 13].

The leverage approximation algorithm of [8] uses a SRHT, $\mathbf{\Pi}_1 \in \mathbb{R}^{r_1 \times n}$ to first compute the approximate SVD of $\mathbf{X}$,

$\mathbf{\Pi}_1 \mathbf{X} = \mathbf{U}_{\Pi X} \mathbf{\Sigma}_{\Pi X} \mathbf{V}_{\Pi X}^{\top}$. Followed by a second SHRT $\mathbf{\Pi}_2 \in \mathbb{R}^{p \times r_2}$ to compute an approximate orthogonal basis for $\mathbf{X}$

$$\mathbf{R}^{-1} = \mathbf{V}_{\Pi X}\mathbf{\Sigma}_{\Pi X}^{-1} \in \mathbb{R}^{p \times p}, \ \ \tilde{\mathbf{U}} = \mathbf{X}\mathbf{R}^{-1}\mathbf{\Pi}_2 \in \mathbb{R}^{n \times r_2}. \tag{5}$$

The approximate leverage scores are now the squared row norms of $\tilde{\mathbf{U}}$, $\tilde{l}_i = \|\tilde{\mathbf{U}}_i\|^2$.

From [14] we derive the following result relating to randomized approximation of the leverage

$$\tilde{l}_i \leq (1 + \rho_l)l_i \,, \tag{6}$$

where the approximation error, $\rho_l$ depends on the choice of projection dimensions $r_1$ and $r_2$.

The leverage weighted least squares (LEV-LS) algorithm samples rows of $\mathbf{X}$ and $\mathbf{y}$ with probability proportional to $l_i$ (or $\tilde{l}_i$ in the approximate case) and performs least squares on this subsample. The residual error resulting from the leverage weighted least squares is bounded by Eq. (4) implying that LEV-LS and SRHT-LS are equivalent [14]. It is important to note that under the corrupted observation model these approximations will be biased.

## 5  Influence weighted subsampling

In the corrupted observation model, OLS and therefore the random approximations to OLS described in §4 obtain poor predictions. To remedy this, we propose influence weighted subsampling (IWS-LS) which is described in Algorithm 1. IWS-LS subsamples points according to the distribution, $P_i = c/d_i$ where $c$ is a normalizing constant so that $\sum_{i=1}^{n} P_i = 1$. OLS is then estimated on the subsampled points. The sampling procedure ensures that points with high influence are selected infrequently and so the resulting estimate is less biased than the full OLS solution. Several approaches similar in spirit have previously been proposed based on identifying and down-weighting the effect of highly influential observations [19].

Obviously, IWS-LS is impractical in the scenarios we consider since it requires the OLS residuals and full leverage scores. However, we use this as a baseline and to simplify the analysis. In the next section, we propose an approximate influence weighted subsampling algorithm which combines the approximate leverage computation of [8] and the randomized least squares approach of [14].

---

**Algorithm 1** Influence weighted subsampling (IWS-LS).

**Input:** Data: $\mathbf{Z}, \mathbf{y}$

1: ***Solve*** $\widehat{\boldsymbol{\beta}}_{\text{OLS}} = \arg\min_{\boldsymbol{\beta}} \|\mathbf{y} - \mathbf{Z}\boldsymbol{\beta}\|^2$
2: **for** $i = 1 \ldots n$ **do**
3: $\quad e_i = y_i - \mathbf{z}_i \widehat{\boldsymbol{\beta}}_{\text{OLS}}$
4: $\quad l_i = \mathbf{z}_i^\top (\mathbf{Z}^\top \mathbf{Z})^{-1} \mathbf{z}_i$
5: $\quad d_i = e_i^2 l_i / (1 - l_i)^2$
6: **end for**
7: ***Sample rows*** $(\tilde{\mathbf{Z}}, \tilde{\mathbf{y}})$ *of* $(\mathbf{Z}, \mathbf{y})$ *proportional to* $\frac{1}{d_i}$
8: ***Solve*** $\widehat{\boldsymbol{\beta}}_{\text{IWS}} = \arg\min_{\boldsymbol{\beta}} \|\tilde{\mathbf{y}} - \tilde{\mathbf{Z}}\boldsymbol{\beta}\|^2$

**Output:** $\widehat{\boldsymbol{\beta}}_{\text{IWS}}$

---

**Algorithm 2** Residual weighted subsampling (aRWS-LS)

**Input:** Data: $\mathbf{Z}, \mathbf{y}$

1: ***Solve*** $\widehat{\boldsymbol{\beta}}_{SRHT} = \arg\min_{\boldsymbol{\beta}} \|\mathbf{\Pi} \cdot (\mathbf{y} - \mathbf{Z}\boldsymbol{\beta})\|^2$
2: ***Estimate residuals:*** $\tilde{\mathbf{e}} = \mathbf{y} - \mathbf{Z}\widehat{\boldsymbol{\beta}}_{SRHT}$
3: ***Sample rows*** $(\tilde{\mathbf{Z}}, \tilde{\mathbf{y}})$ *of* $(\mathbf{Z}, \mathbf{y})$ *proportional to* $\frac{1}{\tilde{e}_i^2}$
4: ***Solve*** $\widehat{\boldsymbol{\beta}}_{RWS} = \arg\min_{\boldsymbol{\beta}} \|\tilde{\mathbf{y}} - \tilde{\mathbf{Z}}\boldsymbol{\beta}\|^2$

**Output:** $\widehat{\boldsymbol{\beta}}_{RWS}$

---

**Randomized approximation algorithms.**  Using the ideas from §4 and §4 we obtain the following randomized approximation to the influence scores

$$\tilde{d}_i = \frac{\tilde{e}_i^2 \tilde{l}_i}{(1 - \tilde{l}_i)^2}, \tag{7}$$

where $\tilde{e}_i$ is the $i^{th}$ residual error computed using the SRHT-LS estimator. Since the approximation errors of $\tilde{e}_i$ and $\tilde{l}_i$ are bounded (inequalities (4) and (6)), this suggests that our randomized approximation to influence is close to the true influence.

**Basic approximation.**  The first approximation algorithm is identical to Algorithm 1 except that leverage and residuals are replaced by their randomized approximations as in Eq. (7). We refer to this algorithm as Approximate influence weighted subsampling (aIWS-LS). Full details are given in Algorithm 3 in §SI.2.

**Residual Weighted Sampling.** Leverage scores are typically uniform [7, 13] for sub-Gaussian data. Even in the corrupted setting, the difference in leverage scores between corrupted and non-corrupted points is small (see §6). Therefore, the main contribution to the influence for each point will originate from the residual error, $e_i^2$. Consequently, we propose sampling with probability inversely proportional to the approximate residual, $\frac{1}{\hat{e}_i^2}$. The resulting algorithm Residual Weighted Subsampling (`aRWS-LS`) is detailed in Algorithm 2. Although `aRWS-LS` is not guaranteed to be a good approximation to `IWS-LS`, empirical results suggests that it works well in practise and is faster to compute than `aIWS-LS`.

**Computational complexity.** Clearly, the computational complexity of `IWS-LS` is $O\left(np^2\right)$. The computation complexity of `aIWS-LS` is $O\left(np\log n_{subs} + npr_2 + n_{subs}p^2\right)$, where the first term is the cost of `SRHT-LS`, the second term is the cost of approximate leverage computation and the last term solves OLS on the subsampled dataset. Here, $r_2$ is the dimension of the random projection detailed in Eq. (5). The cost of `aRWS-LS` is $O\left(np\log n_{subs} + np + n_{subs}p^2\right)$ where the first term is the cost of `SRHT-LS`, the second term is the cost of computing the residuals **e**, and the last term solves OLS on the subsampled dataset. This computation can be reduced to $O\left(np\log n_{subs} + n_{subs}p^2\right)$. Therefore the cost of both `aIWS-LS` and `aRWS-LS` is $o(np^2)$.

## 6   Estimation error

In this section we will prove an upper bound on the estimation error of `IWS-LS` in the corrupted model. First, we show that the OLS error consists of two additional variance terms that depend on the size and proportion of the corruptions and an additional bias term. We then show that `IWS-LS` can significantly reduce the relative variance and bias in this setting, so that it no longer depends on the magnitude of the corruptions but only on their proportion. We compare these results to recent results from [4, 12] suggesting that consistent estimation requires knowledge about $\Sigma_w$. More recently, [5] show that incomplete knowledge about this quantity results in a biased estimator where the bias is proportional to the uncertainty about $\Sigma_w$. We see that the form of our bound matches these results.

Inequalities are said to hold *with high probability (w.h.p.)* if the probability of failure is not more than $C_1\exp(-C_2\log p)$ where $C_1, C_2$ are positive constants that do not depend on the scaling quantities $n, p, \sigma_w$. The symbol $\lesssim$ means that we ignore constants that do not depend on these scaling quantities. Proofs are provided in the supplement. Unless otherwise stated, $\|\cdot\|$ denotes the $\ell_2$ norm for vectors and the spectral norm for matrices.

**Corrupted observation model.** As a baseline, we first investigate the behaviour of the OLS estimator in the corrupted model.

**Theorem 1** (A bound on $\|\widehat{\boldsymbol{\beta}}_{OLS} - \boldsymbol{\beta}\|$). *If $n \gtrsim \frac{\sigma_x^2\sigma_w^2}{\lambda_{\min}(\Sigma_x)}p\log p$ then w.h.p.*

$$\|\widehat{\boldsymbol{\beta}}_{OLS} - \boldsymbol{\beta}\| \lesssim \left(\left(\sigma_\epsilon\sigma_x + \pi\sigma_\epsilon\sigma_w + \pi\left(\sigma_w^2 + \sigma_w\sigma_x\right)\|\boldsymbol{\beta}\|\right)\sqrt{\frac{p\log p}{n}} + \pi\sigma_w^2\sqrt{p}\|\boldsymbol{\beta}\|\right)\cdot\frac{1}{\lambda} \quad (8)$$

*where $0 < \lambda \le \lambda_{\min}(\Sigma_x) + \pi\lambda_{\min}(\Sigma_w)$.*

**Remark 1** (No corruptions case). *Notice for a fixed $\sigma_w$, taking $\lim_{\pi\to 0}$ or for a fixed $\pi$ taking $\lim_{\sigma_w\to 0}$ (i.e. there are no corruptions) the above error reduces to the least squares result (see for example [4]).*

**Remark 2** (Variance and Bias). *The first three terms in (8) scale with $\sqrt{1/n}$ so as $n \to \infty$, these terms tend towards 0. The last term does not depend on $\sqrt{1/n}$ and so for some non-zero $\pi$ the least squares estimate will incur some bias depending on the fraction and magnitude of corruptions.*

We are now ready to state our theorem characterising the mean squared error of the influence weighted subsampling estimator.

**Theorem 2** (Influence sampling in the corrupted model). *For $n \gtrsim \frac{\sigma_x^2\sigma_w^2}{\lambda_{\min}(\Sigma_{\Theta x})}p\log p$ we have*

$$\|\widehat{\boldsymbol{\beta}}_{IWS} - \boldsymbol{\beta}\| \lesssim \left(\left(\sigma_\epsilon\sigma_x + \frac{\pi\sigma_\epsilon}{(\sigma_w+1)} + \pi\|\boldsymbol{\beta}\|\right)\sqrt{\frac{p\log p}{n_{subs}}} + \pi\sqrt{p}\|\boldsymbol{\beta}\|\right)\cdot\frac{1}{\lambda}$$

*where $0 < \lambda \le \lambda_{\min}(\Sigma_{\Theta x})$ and $\Sigma_{\Theta x}$ is the covariance of the influence weighted subsampled data.*

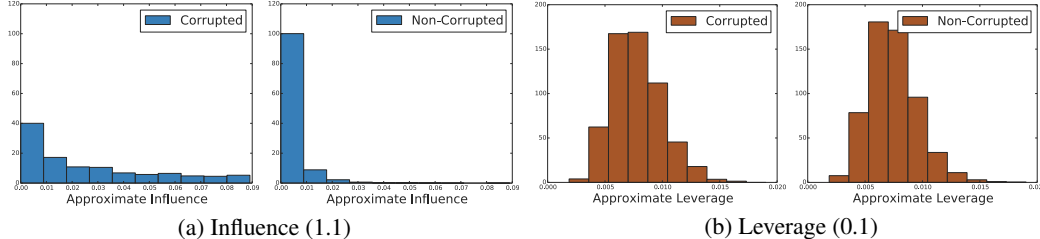

|                          |                           |                        |                            |
| (a) Influence (1.1)      |                           | (b) Leverage (0.1)     |                            |

Figure 1: Comparison of the distribution of the influence and leverage for corrupted and non-corrupted points. The $\ell_1$ distance between the histograms is shown in brackets.

**Remark 3.** *Theorem 2 states that the influence weighted subsampling estimator removes the proportional dependance of the error on $\sigma_w$ so the additional variance terms scale as $O(\pi/\sigma_w \cdot \sqrt{p/n_{subs}})$ and $O(\pi\sqrt{p/n_{subs}})$. The relative contribution of the bias term is $\pi\sqrt{p}\|\boldsymbol{\beta}\|$ compared with $\pi\sigma_w^2\sqrt{p}\|\boldsymbol{\beta}\|$ for the OLS or non-influence-based subsampling methods.*

**Comparison with fully corrupted setting.** We note that the bound in Theorem 1 is similar to the bound in [5] for an estimator where all data points are corrupted (i.e. $\pi = 1$) and where incomplete knowledge of the covariance matrix of the corruptions, $\Sigma_w$ is used. The additional bias in the estimator is proportional to the uncertainty in the estimate of $\Sigma_w$ – in Theorem 1 this corresponds to $\sigma_w^2$. Unbiased estimation is possible if $\Sigma_w$ is known. See the Supplementary Information for further discussion, where the relevant results from [5] are provided in Section SI.6.1 as Lemma 16.

## 7   Experimental results

We compare `IWS-LS` against the methods `SRHT-LS` [14], `ULURU` [7]. These competing methods represent current state-of-the-art in fast randomized least squares. Since `SRHT-LS` is equivalent to `LEV-LS` [9] the comparison will highlight the difference between importance sampling according to the two difference types of regression diagnostic in the corrupted model. Similar to `IWS-LS`, `ULURU` is also a two-step procedure where the first is equivalent to `SRHT-LS`. The second reduces bias by subtracting the result of regressing onto the residual. The experiments with the corrupted data model will demonstrate the difference in robustness of `IWS-LS` and `ULURU` to corruptions in the observations. Note that we do not compare with SGD. Although SGD has excellent properties for large-scale linear regression, we are not aware of a convex loss function which is robust to the corruption model we propose.

We assess the empirical performance of our method compared with standard and state-of-the-art randomized approaches to linear regression in several difference scenarios. We evaluate these methods on the basis of the estimation error: the $\ell_2$ norm of the difference between the true weights and the learned weights, $\|\widehat{\boldsymbol{\beta}} - \boldsymbol{\beta}\|$. We present additional results for root mean squared prediction error (RMSE) on the test set in §SI.7.

For all the experiments on simulated data sets we use $n_{train} = 100,000$, $n_{test} = 1000$, $p = 500$. For datasets of this size, computing exact leverage is impractical and so we report on results for `IWS-LS` in §SI.7. For `aIWS-LS` and `aRWS-LS` we used the same number of sub-samples to approximate the leverage scores and residuals as for solving the regression. For `aIWS-LS` we set $r_2 = p/2$ (see Eq. (5)). The results are averaged over 100 runs.

**Corrupted data.** We investigate the corrupted data noise model described in Eqs. (1)-(2). We show three scenarios where $\pi = \{0.05, 0.1, 0.3\}$. $\mathbf{X}$ and $\mathbf{W}$ were sampled from independent, zero-mean Gaussians with standard deviation $\sigma_x = 1$ and $\sigma_w = 0.4$ respectively. The true regression coefficients, $\boldsymbol{\beta}$ were sampled from a standard Gaussian. We added i.i.d. zero-mean Gaussian noise with standard deviation $\sigma_e = 0.1$.

Figure 1 shows the difference in distribution of influence and leverage between non-corrupted points (top) and corrupted points (bottom) for a dataset with 30% corrupted points. The distribution of leverage is very similar between the corrupted and non-corrupted points, as quantified by the $\ell_1$ difference. This suggests that leverage alone cannot be used to identify corrupted points.

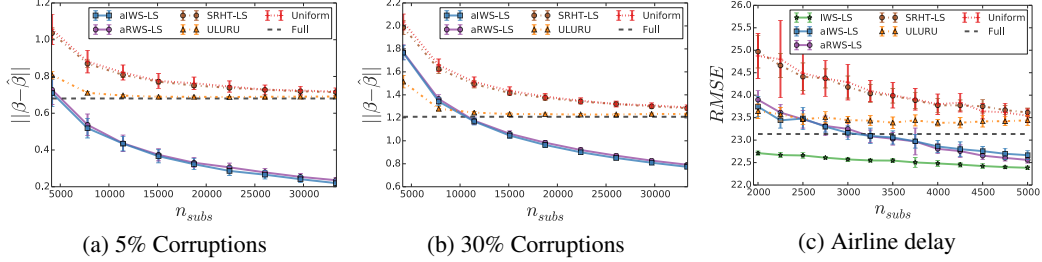

|  (a) 5% Corruptions | (b) 30% Corruptions | (c) Airline delay |

Figure 2: Comparison of mean estimation error and standard deviation on two corrupted simulated datasets and the airline delay dataset.

On the other hand, although there are some corrupted points with small influence, they typically have a much larger influence than non-corrupted points. We give a theoretical explanation of this phenomenon in §SI.3 (remarks 4 and 5).

Figure 2(a) and (b) shows the estimation error and the mean squared prediction error for different subsample sizes. In this setting, computing `IWS-LS` is impractical (due to the exact leverage computation) so we omit the results but we notice that `aIWS-LS` and `aRWS-LS` quickly improve over the full least squares solution and the other randomized approximations in all simulation settings. In all cases, influence based methods also achieve lower-variance estimates.

For $30\%$ corruptions for a small number of samples `ULURU` outperforms the other subsampling methods. However, as the number of samples increases, influence based methods start to outperform OLS. Here, `ULURU` converges quickly to the OLS solution but is not able to overcome the bias introduced by the corrupted datapoints. Results for $10\%$ corruptions are shown in Figs. 5 and 6 and we provide results on smaller corrupted datasets (to show the performance of `IWS-LS`) as well as non-corrupted data simulated according to [13] in §SI.7.

**Airline delay dataset** The dataset consists of details of all commercial flights in the USA over 20 years. Dataset along with visualisations available from `http://stat-computing.org/dataexpo/2009/`. Selecting the first $n_{train} = 13,000$ US Airways flights from January 2000 (corresponding to approximately 1.5 weeks) our goal is to predict the delay time of the next $n_{test} = 5,000$ US Airways flights. The features in this dataset consist of a binary vector representing origin-destination pairs and a real value representing distance ($p = 170$).

The dataset might be expected to violate the usual i.i.d. sub-Gaussian design assumption of standard linear regression since the length of delays are often very different depending on the day. For example, delays may be longer due to public holidays or on weekends. Of course, such regular events could be accounted for in the modelling step, but some unpredictable outliers such as weather delay may also occur. Results are presented in Figure 2(c), the RMSE is the error in predicted delay time in minutes. Since the dataset is smaller, we can run `IWS-LS` to observe the accuracy of `aIWS-LS` and `aRWS-LS` in comparison. For more than 3000 samples, these algorithm outperform OLS and quickly approach `IWS-LS`. The result suggests that the corrupted observation model is a good model for this dataset. Furthermore, `ULURU` is unable to achieve the full accuracy of the OLS solution.

# 8 Conclusions

We have demonstrated theoretically and empirically under the generalised corrupted observation model that influence weighted subsampling is able to significantly reduce both the bias and variance compared with the OLS estimator and other randomized approximations which do not take influence into account. Importantly our fast approximation, `aRWS-LS` performs similarly to `IWS-LS`. We find `ULURU` quickly converges to the OLS estimate, although it is not able to overcome the bias induced by the corrupted datapoints despite its two-step procedure. The performance of `IWS-LS` relative to OLS in the airline delay problem suggests that the corrupted observation model is a more realistic modelling scenario than the standard sub-Gaussian design model for some tasks. Software is available at `http://people.inf.ethz.ch/kgabriel/software.html`.

**Acknowledgements.** We thank David Balduzzi, Cheng Soon Ong and the anonymous reviewers for invaluable discussions, suggestions and comments.

## Footnotes

[2]Unlike [5, 12] and others we do not consider sparsity in our solution since $n \gg p$.

[3]The expression we use is also called *Cook's distance* [2].

[4] For the Hadamard transform, $n$ must be a power of two but other transforms exist (e.g. DCT, DFT) for which similar theoretical guarantees hold and there is no restriction on $n$.

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
