[Supplementary Material]

# Supplementary Information for Fast and Robust Least Squares Estimation in Corrupted Linear Models

Here we collect supplementary technical details, discussion and empirical results which support the results presented in the main text.

## SI.1 Software

We have made available a software package available for Python which implements

- `IWS-LS`,
- `aIWS-LS` and
- `aRWS-LS`,

along with the methods we compare against

- `SRHT-LS` and
- `ULURU`.

The software is available at `http://people.inf.ethz.ch/kgabriel/software.html`.

## SI.2 Approximate Influence Weighted Algorithm

Here we present a detailed description of the approximate influence weighted subsampling (`aIWS-LS`) algorithm. Steps 2, 3 and 4 are required for the approximate leverage computation. Step 3 could be replaced with the QR decomposition.

---

**Algorithm 3** Approximate influence weighted subsampling (`aIWS-LS`).

---

**Input:** Data: $\mathbf{Z}, \mathbf{y}$

1: **Solve** $\widehat{\boldsymbol{\beta}}_{SRHT} = \arg\min_{\boldsymbol{\beta}} \|\boldsymbol{\Pi}_1 \cdot \mathbf{y} - \boldsymbol{\Pi}_1 \cdot \mathbf{Z}\boldsymbol{\beta}\|^2$
2: **SVD:** $(\mathbf{U}, \boldsymbol{\Sigma}, \mathbf{V}) = \boldsymbol{\Pi}_1 \cdot \mathbf{Z}$ {Compute basis for randomized leverage approximation.}
3: $\mathbf{R}^{-1} = \mathbf{V}\boldsymbol{\Sigma}^{-1}$
4: $\tilde{\mathbf{U}} = \mathbf{Z}\mathbf{R}^{-1} \cdot \boldsymbol{\Pi}_2$
5: **for** $i = 1 \ldots n$ **do**
6: $\quad \tilde{l}_i = \|\tilde{\mathbf{U}}_i\|$
7: $\quad \tilde{e}_i = y_i - \mathbf{z}_i\widehat{\boldsymbol{\beta}}_{SRHT}$
8: $\quad \tilde{d}_i = \tilde{e}_i^2 \tilde{l}_i / (1 - \tilde{l}_i)^2$
9: **end for**
10: **Sample rows $(\tilde{\mathbf{Z}}, \tilde{\mathbf{y}})$ of $(\mathbf{Z}, \mathbf{y})$ proportional to** $\frac{1}{\tilde{d}_i}$
11: **Solve** $\widehat{\boldsymbol{\beta}}_{aIWS} = \arg\min_{\boldsymbol{\beta}} \|\tilde{\mathbf{y}} - \tilde{\mathbf{Z}}\boldsymbol{\beta}\|^2$

**Output:** $\widehat{\boldsymbol{\beta}}_{aIWS}$

---

## SI.3 Leverage and Influence

Here we provide detailed derivations of leverage and influence terms as well as the full statement and proofs of finite sample bounds under the sub-Gaussian design and corrupted design models which are abbreviated in the main text as Lemmas 5, 6, 7, and 8.

Here we provide a full derivation of the leave-one-out estimator of $\widehat{\boldsymbol{\beta}}$ which appears in less detail in [2].

**Proposition 3** (Derivation of $\widehat{\boldsymbol{\beta}}_{-i}$). *Defining* $e_i = \hat{y}_i - y_i$ *and* $\boldsymbol{\Sigma} = \mathbf{X}^\top \mathbf{X}$

$$
\begin{aligned}
\widehat{\boldsymbol{\beta}}_{-i} &= \left(\boldsymbol{\Sigma} - \mathbf{x}_i^\top \mathbf{x}_i\right)^{-1} \left(\mathbf{X}^\top \mathbf{y} - \mathbf{x}_i^\top y_i\right) \\
&= \left(\boldsymbol{\Sigma}^{-1} + \frac{\boldsymbol{\Sigma}^{-1} \mathbf{x}_i \mathbf{x}_i^\top \boldsymbol{\Sigma}^{-1}}{1 - l_i}\right) \left(\mathbf{X}^\top \mathbf{y} - \mathbf{x}_i^\top y_i\right) \\
&= \widehat{\boldsymbol{\beta}} - \boldsymbol{\Sigma}^{-1} \mathbf{x}_i^\top \left(y_i + \frac{\mathbf{x}_i \boldsymbol{\Sigma}^{-1} \mathbf{X} \mathbf{y} - \mathbf{x}_i \boldsymbol{\Sigma}^{-1} \mathbf{x}_i^\top y_i}{1 - l_i}\right) \\
&= \widehat{\boldsymbol{\beta}} - \boldsymbol{\Sigma}^{-1} \mathbf{x}_i^\top \left(y_i + \frac{\hat{y}_i - l_i y_i}{1 - l_i}\right) \\
&= \widehat{\boldsymbol{\beta}} - \boldsymbol{\Sigma}^{-1} \mathbf{x}_i^\top \left(y_i + \frac{e_i}{1 - l_i} - \frac{y_i(1 - l_i)}{1 - l_i}\right) \\
&= \widehat{\boldsymbol{\beta}} - \frac{\boldsymbol{\Sigma}^{-1} \mathbf{x}_i^\top e_i}{1 - l_i}
\end{aligned}
$$

*Where the first equality comes from a straightforward application of the Sherman Morrison formula.*

Here we provide a derivation of the leave-one-out estimator in the corrupted model where the point we removed is corrupted.

**Proposition 4** (Derivation of $\widehat{\boldsymbol{\beta}}_{-m}$). *By proposition 3. Defining*

$$
e_m = \hat{y}_m - y_m = (\mathbf{x}_m + \mathbf{w}_m)\widehat{\boldsymbol{\beta}} - y_m \quad and
$$

$$
l_m = (\mathbf{x}_m + \mathbf{w}_m)\boldsymbol{\Sigma}^{-1}(\mathbf{x}_m + \mathbf{w}_m)^\top
$$

*where* $\boldsymbol{\Sigma} = \mathbf{Z}^\top \mathbf{Z}$, *we have that*

$$
\widehat{\boldsymbol{\beta}}_{-m} = \widehat{\boldsymbol{\beta}} - \frac{\boldsymbol{\Sigma}^{-1} (\mathbf{x}_m + \mathbf{w}_m)^\top e_m}{1 - l_m}.
$$

### SI.3.1 Results for Sub-Gaussian random design

**Lemma 5** (Leverage). *The leverage of a non-corrupted point is bounded by*

$$
l_i \leq \sigma_x^2 \cdot O\left((p/\sqrt{n})^2\right) \tag{9}
$$

*where the exact form of the* $O\left((p/\sqrt{n})^2\right)$ *term is given in the supplementary material.*

**Lemma 6** (Influence). *Defining* $E := \|\widehat{\boldsymbol{\beta}}_{OLS} - \boldsymbol{\beta}\|$, *the influence of a non-corrupted point is*

$$
d_i \leq C_i \left(\sigma_x \sigma_\epsilon + \sigma_x^2 E\right). \tag{10}
$$

*The* $C_i$ *term is proportional to* $\log p \sqrt{p} \|\boldsymbol{\Sigma}^{-1}\| / (1 - l_i)$.

**Proof of Lemma 5.** Lemma 5 states

$$
l_i \leq \sigma_x^2 \cdot \left(\frac{p + 2\log p + 2\sqrt{p \log p}}{\sqrt{n} - C\sqrt{p} - \sqrt{\log p}}\right)^2.
$$

From the Eigen-decomposition, $\boldsymbol{\Sigma} = \mathbf{V} \boldsymbol{\Lambda} \mathbf{V}^\top$. Define $\mathbf{A} = \boldsymbol{\Lambda}^{-1/2} \mathbf{V}$ such that $\mathbf{A}^\top \mathbf{A} = \boldsymbol{\Sigma}^{-1}$. We have

$$
\begin{aligned}
l_i &= \mathbf{x}_i \boldsymbol{\Sigma}^{-1} \mathbf{x}_i^\top \\
&= \|\mathbf{A} \mathbf{x}_i^\top\|^2
\end{aligned}
$$

Since $\mathbf{x}$ and $\mathbf{w}$ are sub-Gaussian random vectors so the above quadratic form is bounded by Lemma 14, setting the parameter $t = \log p$. We combine this with the following inequalities

$$
\sqrt{\operatorname{tr}\left(\boldsymbol{\Sigma}^{-2}\right)} = \|\boldsymbol{\Sigma}^{-1}\|_F \leq \sqrt{p} \|\boldsymbol{\Sigma}^{-1}\| = \sqrt{p} \sigma_1(\mathbf{A})^2
$$

and
$$\mathrm{tr}\left(\mathbf{\Sigma}^{-1}\right) = \|\mathbf{A}\|_F^2 \leq (\sqrt{p}\|\mathbf{A}\|)^2 = p\sigma_1(\mathbf{A})^2$$
which relate the Frobenius norm with the spectral norm. We also make use of the relationship $\sigma_n(\mathbf{Z})^{-1} = \sigma_1(\mathbf{A})$ where $\mathbf{Z} = \mathbf{X} + \mathbf{W}$ to obtain
$$\|\mathbf{A}\mathbf{x}\|^2 \leq \sigma_x^2 \sigma_n(\mathbf{Z})^{-2}\left(p + 2\log p + 2\sqrt{p\log p}\right)$$
which holds with high probability.

In order for this bound to not be vacuous in our application, it must be smaller than 1. In order to ensure this, we need bound $\sigma_n(\mathbf{Z})^{-1}$ using Lemma 15 and setting $\tau = \sqrt{c_0 \log p}$ to obtain the following which holds with high probability
$$\|\mathbf{A}\mathbf{x}\|^2 \leq \sigma_x^2 \left(\frac{p + 2\log p + 2\sqrt{p\log p}}{\sqrt{n} - C\sqrt{p} - \tau}\right)^2$$
$$\leq \sigma_x^2 \left(\frac{p + 2\log p + 2\sqrt{p\log p}}{\sqrt{n} - C\sqrt{p} - \sqrt{\log p}}\right)^2.$$
$\square$

**Proof of Lemma 6.**  Defining $\mathbf{\Sigma} = \mathbf{Z}^\top \mathbf{Z}$, Lemma 6 states
$$\|\widehat{\boldsymbol{\beta}}_{-i} - \widehat{\boldsymbol{\beta}}\| \leq \frac{\|\mathbf{\Sigma}^{-1}\|}{1 - l_i}\left(\sigma_x \sigma_\epsilon + 2\sigma_x^2\|\boldsymbol{\beta} - \widehat{\boldsymbol{\beta}}\|\right)\sqrt{p}\log p.$$
Using Proposition 3 we have
$$\begin{aligned}
\|\widehat{\boldsymbol{\beta}}_{-i} - \widehat{\boldsymbol{\beta}}\| &= \frac{1}{1 - l_i}\|\mathbf{\Sigma}^{-1}\mathbf{x}_i^\top e_i\| \\
&= \frac{1}{1 - l_i}\|\mathbf{\Sigma}^{-1}\mathbf{x}_i^\top\left(\epsilon + \mathbf{x}_i\left(\boldsymbol{\beta} - \widehat{\boldsymbol{\beta}}\right)\right)\| \\
&\leq \frac{1}{1 - l_i}\|\mathbf{\Sigma}^{-1}\|\|\mathbf{x}_i^\top\epsilon + \mathbf{x}_i^\top\mathbf{x}_i\left(\boldsymbol{\beta} - \widehat{\boldsymbol{\beta}}\right)\| \\
&\leq \frac{1}{1 - l_i}\|\mathbf{\Sigma}^{-1}\|\left(\|\mathbf{x}_i^\top\epsilon\| + \|\mathbf{x}_i^\top\mathbf{x}_i(\boldsymbol{\beta} - \widehat{\boldsymbol{\beta}})\|\right).
\end{aligned}$$
Using Corollary 12 to bound $\|\mathbf{x}_i^\top\epsilon\|$ and $\|\mathbf{x}_i^\top\mathbf{x}_i(\boldsymbol{\beta} - \widehat{\boldsymbol{\beta}})\|$ (since for these terms $n = 1$ and so Lemma 11 does not immediately apply) completes the proof. $\square$

### SI.3.2  Results for corrupted observations

**Lemma 7** (Leverage of corrupted point). *The leverage of a corrupted point is bounded by*
$$l_m \leq (\sigma_x^2 + \sigma_w^2) \cdot O\left((p/\sqrt{n})^2\right). \tag{11}$$

**Remark 4** (Comparison of leverage). *Comparing this with Eq. (9), when $n$ is large, the dominant term is $O((p/\sqrt{n})^2)$ which implies that the difference in leverage between a corrupted and non-corrupted point – particularly when the magnitude of corruptions is not large – is small. This suggests that it may not be possible to distinguish between the corrupted and non-corrupted points by only comparing leverage scores.*

**Lemma 8** (Influence of corrupted point). *Defining $E := \|\widehat{\boldsymbol{\beta}}_{OLS} - \boldsymbol{\beta}\|$, the influence of a corrupted point is*
$$\begin{aligned}
d_m \leq &C_m(\sigma_x\sigma_w + \sigma_w^2)\|\boldsymbol{\beta}\| + (\sigma_x^2 + 2\sigma_x\sigma_w + \sigma_w^2)E \\
&+ (\sigma_x + \sigma_w)\sigma_\epsilon.
\end{aligned} \tag{12}$$

**Remark 5** (Comparison of influence). *Here, $C_m$ differs from $C_i$ in Lemma 6 only in its dependence on the leverage of a corrupted instead of non-corrupted point and so for large $n$, $C_i \approx C_m$. It can be seen that the influence of the corrupted point includes a bias term similar to the one which appears in Eq. (8). This suggests that the relative difference between the influence of a non-corrupted and corrupted point will be larger than the respective relative difference in leverage. All of the information relating to the proportion of corrupted points is contained within $E$.*

**Proof of Lemma 7.**   Lemma 7 states

$$l_m \leq (\sigma_x^2 + \sigma_w^2) \cdot \left( \frac{p + 2\log p + 2\sqrt{p\log p}}{\sqrt{n} - C\sqrt{p} - \sqrt{\log p}} \right)^2 .$$

The proof follows from rewriting $l_m = \|\mathbf{A}(\mathbf{x}_m + \mathbf{w}_m)^\top\|^2$ and following the same steps as the proof of Lemma 5 above. $\square$

**Proof of Lemma 8.**   Lemma 8 states

$$\|\widehat{\boldsymbol{\beta}}_{-m} - \widehat{\boldsymbol{\beta}}\| \leq \frac{\|\boldsymbol{\Sigma}^{-1}\|}{1 - l_m} \left( 2(\sigma_x\sigma_w + \sigma_w^2)\|\boldsymbol{\beta}\| + 2(\sigma_x^2 + \sigma_x\sigma_w + \sigma_w^2) \cdot \|\boldsymbol{\beta} - \widehat{\boldsymbol{\beta}}\| + 2(\sigma_x + \sigma_w)\sigma_\epsilon \right)$$
$$\cdot \sqrt{p}\log p.$$

From Proposition 4 and following the same argument as Lemma 6 we have

$$
\begin{aligned}
\|\widehat{\boldsymbol{\beta}}_{-m} - \widehat{\boldsymbol{\beta}}\| =& \frac{1}{1 - l_m}\|\boldsymbol{\Sigma}^{-1}(\mathbf{x}_m + \mathbf{w}_m)^\top e_m\| \\
\leq& \frac{1}{1 - l_m}\|\boldsymbol{\Sigma}^{-1}(\mathbf{x}_m + \mathbf{w}_m)^\top \left( (\mathbf{x}_m + \mathbf{w}_m)(\boldsymbol{\beta} - \widehat{\boldsymbol{\beta}}) + \mathbf{w}_m\boldsymbol{\beta} + \epsilon \right)\| \\
\leq& \frac{1}{1 - l_m}\|\boldsymbol{\Sigma}^{-1}\| \left( \|\mathbf{x}_m^\top \mathbf{w}_m \boldsymbol{\beta}\| + \|\mathbf{w}_m^\top \mathbf{w}_m \boldsymbol{\beta}\| + \|\mathbf{x}_m^\top \epsilon\| + \|\mathbf{w}_m^\top \epsilon\| \right. \\
& \left. + \|\left(\mathbf{x}_m^\top \mathbf{x}_m + \mathbf{w}_m^\top \mathbf{w}_m + 2\mathbf{x}_m^\top \mathbf{w}_m\right)(\boldsymbol{\beta} - \widehat{\boldsymbol{\beta}})\| \right).
\end{aligned}
$$

Applying the triangle inequality followed by Corollary 12 and noting that $(\sigma_x\sigma_w + 2\sigma_w^2) \leq 2(\sigma_x\sigma_w + \sigma_w^2)$ completes the proof. $\square$

## SI.4   Estimation error in sub-Gaussian model

Using the definition of influence above, we can state the following theorem characterising the error of the influence weighted subsampling estimator in the sub-Gaussian design setting.

**Theorem 9** (Sub-gaussian design influence weighted subsampling)**.**  *Defining* $E = \|\widehat{\boldsymbol{\beta}}_{OLS} - \boldsymbol{\beta}\|$ *for* $n \gtrsim \frac{\sigma_x^2}{\lambda_{\min}(\Sigma_{\Theta x})}p\log p$ *we have*

$$\|\widehat{\boldsymbol{\beta}}_{IWS} - \boldsymbol{\beta}\| \lesssim \frac{1}{\lambda} \cdot \frac{\sigma_\epsilon}{\lambda_{\min}(\Sigma_x)(\sigma_\epsilon + 2\sigma_x E)} \cdot \sqrt{\frac{1}{rn}}$$

*where* $0 \leq \lambda \leq \lambda_{\min}(\Sigma_{\Theta x})$ *and* $\Sigma_{\Theta x}$ *is the covariance of the influence weighted subsampled data and* $r = n_{subs}/n$.

**Remark 6.**  *Theorem 9 states that in the non-corrupted sub-Gaussian model, the influence weighted subsampling estimator is consistent. Furthermore, if we set the sampling proportion,* $r \geq O\left(1/p\right)$, *the error scales as* $O\left(\sqrt{p/n}\right)$. *Therefore, similar to* ULURU *there is no dependence on the subsampling proportion.*

## SI.5   Proof of main theorems

In this section we provide proofs of our main theorems which describe the properties of the influence weighted subsampling estimator in the sub-Gaussian random design case, the OLS estimator in the corrupted setting and finally our influence weighted subsampling estimator in the corrupted setting.

In order to prove our results we require the following lemma

**Lemma 10** (A general bound on $\|\widehat{\boldsymbol{\beta}} - \boldsymbol{\beta}\|$ from [5]). *Suppose the following strong convexity condition holds:* $\lambda_{\min}(\widehat{\boldsymbol{\Sigma}}) \geq \lambda > 0$. *Then the estimation error satisfies*

$$\|\widehat{\boldsymbol{\beta}} - \boldsymbol{\beta}\| \lesssim \frac{1}{\lambda}\|\hat{\gamma} - \widehat{\boldsymbol{\Sigma}}\boldsymbol{\beta}\|.$$

*Where $\hat{\gamma}$, $\widehat{\boldsymbol{\Sigma}}$ are estimators for $\mathbb{E}\left[\mathbf{X}^\top \mathbf{y}\right]$ and $\mathbb{E}\left[\mathbf{X}^\top \mathbf{X}\right]$ respectively*

To obtain the results for our method in the non-corrupted and corrupted setting we can simply plug in our specific estimates for $\hat{\gamma}$ and $\widehat{\boldsymbol{\Sigma}}$.

**Proof of Theorem 9.** Through subsampling according to influence, we solve the problem

$$\widehat{\boldsymbol{\beta}}_{\text{IWS}} = \arg\min_{\boldsymbol{\beta}} \|\Theta\mathbf{y} - \Theta\mathbf{X}\boldsymbol{\beta}\|^2$$

where $\Theta = \sqrt{\frac{n}{n_{subs}}}\mathbf{S}\mathbf{D}$. $\mathbf{S}$ is a subsampling matrix, $\mathbf{D}$ is a diagonal matrix whose entries are $\sqrt{P_i/n} = \sqrt{c/d_i n}$ where $c$ is a constant which ensures $\sum_{i=1}^n P_i = 1$.

$$D_{ii}^2 \propto \left(\frac{\|\boldsymbol{\Sigma}^{-1}\|}{1 - l_i}\left(\sigma_x\sigma_\epsilon + 2\sigma_x^2\|\boldsymbol{\beta} - \widehat{\boldsymbol{\beta}}\|\right)\sqrt{p}\log p\right)^{-1}. \tag{13}$$

Setting $\hat{\gamma} = (\Theta\mathbf{X})^\top\mathbf{y}$, $\widehat{\boldsymbol{\Sigma}} = (\Theta\mathbf{X})^\top(\Theta\mathbf{X})$, by Lemma 10 the error of the influence weighted subsampling estimator is given by

$$
\begin{aligned}
\frac{1}{\lambda}\|\hat{\gamma} - \widehat{\boldsymbol{\Sigma}}\boldsymbol{\beta}\| &= \|(\Theta\mathbf{X})^\top(\Theta\mathbf{y}) - (\Theta\mathbf{X})^\top(\Theta\mathbf{X})\boldsymbol{\beta}\| \\
&= \frac{1}{\lambda}\|(\Theta\mathbf{X})^\top(\Theta\epsilon) + (\Theta\mathbf{X})^\top(\Theta\mathbf{X})\boldsymbol{\beta} - (\Theta\mathbf{X})^\top(\Theta\mathbf{X})\boldsymbol{\beta}\| \\
&= \frac{1}{\lambda}\|(\Theta\mathbf{X})^\top(\Theta\epsilon)\| \tag{14}
\end{aligned}
$$

Now, by Lemma 11 we have

$$\|\mathbf{X}^\top\epsilon\| \leq \sigma_x\sigma_\epsilon\sqrt{\frac{p\log p}{n}}$$

and so defining $E = \|\boldsymbol{\beta} - \widehat{\boldsymbol{\beta}}\|$,

$$
\begin{aligned}
\|(\Theta\mathbf{X})^\top\Theta\epsilon\| &\leq \|\frac{1}{rn}\sum_{i=1}^{rn} p_i\| \cdot \|(\mathbf{S}\mathbf{X})^\top\mathbf{S}\epsilon\| \\
&\leq \|\frac{1}{rn}\sum_{i=1}^{rn}(1 - l_i)\|\frac{\sigma_x\sigma_\epsilon\sqrt{p\log p/rn}}{\|\boldsymbol{\Sigma}^{-1}\|\left(\sigma_x\sigma_\epsilon + 2\sigma_x^2 E\right)\sqrt{p}\log p} \\
&\leq \frac{\sigma_x\sigma_\epsilon\sqrt{p\log p/rn}}{\|\boldsymbol{\Sigma}^{-1}\|\left(\sigma_x\sigma_\epsilon + 2\sigma_x^2 E\right)\sqrt{p}\log p} \\
&\leq \frac{\sigma_\epsilon\sqrt{1/rn}}{\lambda_{\min}(\Sigma_x)(\sigma_\epsilon + 2\sigma_x E)} \tag{15}
\end{aligned}
$$

where the third inequaltiy uses the fact that $\sum_{i=1}^n(1 - l_i) \leq n$.

Define $\Sigma_{\Theta x} = \mathbb{E}\left[(\Theta\mathbf{X})^\top(\Theta\mathbf{X})\right]$. Now, when $n \gtrsim \frac{(\sigma_x^2)p\log p}{\lambda_{\min}(\Sigma_{\Theta x})}$ using Lemma 13 with $\lambda = \lambda_{\min}(\Sigma_{\Theta x})$ we have w.h.p. $\lambda_1((\Theta\mathbf{X})^\top(\Theta\mathbf{X}) - \Sigma_{\Theta x}) \leq \frac{1}{54}\lambda_{\min}(\Sigma_{\Theta x})$. It follows that

$$
\begin{aligned}
\lambda_{\min}((\Theta\mathbf{X})^\top(\Theta\mathbf{X})) &= \inf_{\|\mathbf{v}\|=1}\mathbf{v}^\top\left(\Sigma_{\Theta x} + (\Theta\mathbf{X})^\top(\Theta\mathbf{X}) - \Sigma_{\Theta x}\right)\mathbf{v} \\
&\geq \lambda_{\min}(\Sigma_{\Theta x}) - \lambda_1((\Theta\mathbf{X})^\top(\Theta\mathbf{X}) - \Sigma_{\Theta x}) \\
&\geq \frac{1}{2}\lambda_{\min}(\Sigma_{\Theta x}). \tag{16}
\end{aligned}
$$

Using (15) and (15) in Eq. (14) completes the proof.

$\square$

**Remark 7** (Scaling by $\pi$). *In the following, with some abuse of notation we will write $U\mathbf{W}$ as $\mathbf{W}$.
Now,*

$$\|\mathbf{W}\| := \|U\mathbf{W}\|$$
$$\leq \pi\|\mathbf{W}\|.$$

**Proof of Theorem 1.** Setting $\hat{\gamma} = \mathbf{Z}^\top \mathbf{y}$, $\widehat{\Sigma} = \mathbf{Z}^\top \mathbf{Z}$ we have

$$\|\hat{\gamma} - \widehat{\Sigma}\boldsymbol{\beta}\| = \|(\mathbf{X}+\mathbf{W})^\top \mathbf{y} - (\mathbf{X}+\mathbf{W})^\top(\mathbf{X}+\mathbf{W})\boldsymbol{\beta}\|$$
$$= \|\mathbf{X}^\top(\mathbf{X}\boldsymbol{\beta}+\epsilon) + \mathbf{W}^\top(\mathbf{X}\boldsymbol{\beta}+\epsilon) - \mathbf{X}^\top\mathbf{X}\boldsymbol{\beta} - \mathbf{W}^\top\mathbf{W}\boldsymbol{\beta} - \mathbf{X}^\top\mathbf{W}\boldsymbol{\beta} - \mathbf{W}^\top\mathbf{X}\boldsymbol{\beta}\|$$
$$= \|\mathbf{X}^\top\epsilon + \mathbf{W}^\top\epsilon - \mathbf{X}^\top\mathbf{W}\boldsymbol{\beta} - \mathbf{W}^\top\mathbf{W}\boldsymbol{\beta}\|$$
$$\leq \|\mathbf{X}^\top\epsilon\| + \|\mathbf{W}^\top\epsilon\| + \|\mathbf{X}^\top\mathbf{W}\boldsymbol{\beta}\| + \|\mathbf{W}^\top\mathbf{W}\boldsymbol{\beta}\|.$$

From Lemma 11 and Remark 7 we have w.h.p.

$$\|\mathbf{X}^\top\epsilon\| \leq \sigma_x\sigma_\epsilon\sqrt{\frac{p\log p}{n}} \tag{17}$$

$$\|\mathbf{W}^\top\epsilon\| \leq \pi\sigma_w\sigma_\epsilon\sqrt{\frac{p\log p}{n}} \tag{18}$$

$$\|\mathbf{X}^\top\mathbf{W}\boldsymbol{\beta}\| \leq \pi\sigma_x\sigma_w\|\boldsymbol{\beta}\|\sqrt{\frac{p\log p}{n}} \tag{19}$$

$$\|\mathbf{W}^\top\mathbf{W}\boldsymbol{\beta}\| = \|\left(\mathbf{W}^\top\mathbf{W} + \sigma_w^2\mathbf{I}_p - \sigma_w^2\mathbf{I}_p\right)\boldsymbol{\beta}\|$$
$$\leq \|\left(\mathbf{W}^\top\mathbf{W} - \sigma_w^2\mathbf{I}_p\right)\boldsymbol{\beta}\| + \sigma_w^2\|\boldsymbol{\beta}\|$$
$$\leq \pi\sigma_w^2\left(C\sqrt{\frac{p\log p}{n}} + \sqrt{p}\right)\|\boldsymbol{\beta}\|. \tag{20}$$

Now, when $n \gtrsim \frac{(\sigma_x^2\sigma_w^2)p\log p}{\lambda_{\min}(\Sigma_x)}$ using Lemma 13 with $\lambda = \lambda_{\min}(\Sigma_x)$ we have w.h.p. $\lambda_1(\mathbf{Z}^\top\mathbf{Z} - (\Sigma_x + \Sigma_w)) \leq \frac{1}{54}\lambda_{\min}(\Sigma_x)$. It follows that

$$\lambda_{\min}(\mathbf{Z}^\top\mathbf{Z}) = \inf_{\|\mathbf{v}\|=1} \mathbf{v}^\top\left(\Sigma_x + \Sigma_w + \mathbf{Z}^\top\mathbf{Z} - (\Sigma_x + \Sigma_w)\right)\mathbf{v}$$
$$\geq \lambda_{\min}(\Sigma_x) + \lambda_{\min}(\Sigma_w) - \lambda_1(\mathbf{Z}^\top\mathbf{Z} - (\Sigma_x + \Sigma_w))$$
$$\geq \frac{1}{2}\lambda_{\min}(\Sigma_x) + \pi\lambda_{\min}(\Sigma_w).$$

Using Lemma 10 with Eqs. (17-20) and the above bound for $\lambda = \lambda_{\min}(\mathbf{Z}^\top\mathbf{Z})$ completes the proof. $\square$

**Proof of Theorem 2.** when $n \gtrsim \frac{(\sigma_x^2\sigma_w^2)p\log p}{\lambda_{\min}(\Sigma_{\Theta x})}$ using Lemma 13 with $\lambda = \lambda_{\min}(\Sigma_{\Theta x})$ we have w.h.p. $\lambda_1((\Theta\mathbf{Z})^\top(\Theta\mathbf{Z}) - \Sigma_{\Theta x}) \leq \frac{1}{54}\lambda_{\min}(\Sigma_{\Theta x})$. It follows that

$$\lambda_{\min}((\Theta\mathbf{Z})^\top(\Theta\mathbf{Z})) = \inf_{\|\mathbf{v}\|=1} \mathbf{v}^\top\left(\Sigma_{\Theta x} + (\Theta\mathbf{Z})^\top(\Theta\mathbf{Z}) - \Sigma_{\Theta x}\right)\mathbf{v}$$
$$\geq \lambda_{\min}(\Sigma_{\Theta x}) - \lambda_1((\Theta\mathbf{Z})^\top(\Theta\mathbf{Z}) - \Sigma_{\Theta x})$$
$$\geq \frac{1}{2}\lambda_{\min}(\Sigma_{\Theta x}).$$

From the bound in Lemma 10 we have

$$\|\hat{\gamma} - \widehat{\boldsymbol{\Sigma}}\boldsymbol{\beta}\| \leq \|(\Theta\mathbf{X})^\top (\Theta\epsilon)\| + \|(\Theta\mathbf{W})^\top (\Theta\epsilon)\|$$
$$+ \|(\Theta\mathbf{X})^\top (\Theta\mathbf{W})\boldsymbol{\beta}\| + \|(\Theta\mathbf{W})^\top (\Theta\mathbf{W})\boldsymbol{\beta}\|.$$

We now aim to show that the relative contribution of the corrupted points is decreased under the influence weighted subsampling scheme. To show this, we first multiply both corrupted and non-corrupted points by

$$\|\boldsymbol{\Sigma}^{-1}\| \left(\sigma_x\sigma_\epsilon + 2\sigma_x^2\|\boldsymbol{\beta} - \widehat{\boldsymbol{\beta}}\|\right) \log p\sqrt{p}.$$

This is equivalent to multiplying the non-corrupted points by the subsampling matrix $\mathbf{S}$ and scaling and subsampling the corrupted points by the following term $\Theta_M = \sqrt{\frac{n}{n_{subs}}}\mathbf{S}\mathbf{D}_M$ where $\mathbf{D}_M$ has squared diagonal entries proportional to

$$D_M^2 \propto \frac{1}{n} \cdot \frac{\sigma_\epsilon\sigma_x + 2\sigma_x^2 E}{2(\sigma_w^2 + \sigma_w\sigma_x)\|\boldsymbol{\beta}\| + 2(\sigma_w^2 + \sigma_w\sigma_x + \sigma_x^2)E + 2(\sigma_w + \sigma_x)\sigma_\epsilon}.$$

Now we have

$$\|\hat{\gamma} - \widehat{\boldsymbol{\Sigma}}\boldsymbol{\beta}\| \lesssim \|(\mathbf{S}\mathbf{X})^\top (\mathbf{S}\epsilon)\| + \|(\Theta_M\mathbf{W})^\top (\Theta_M\epsilon)\|$$
$$+ \|(\Theta_M\mathbf{X})^\top (\Theta_M\mathbf{W})\boldsymbol{\beta}\| + \|(\Theta_M\mathbf{W})^\top (\Theta_M\mathbf{W})\boldsymbol{\beta}\|.$$

Applying Lemma 11 we have w.h.p.

$$\|(\mathbf{S}\mathbf{X})^\top(\mathbf{S}\epsilon)\| \lesssim \sigma_x\sigma_\epsilon\sqrt{\frac{p\log p}{rn}} \tag{21}$$

$$\|(\Theta_M\mathbf{W})^\top(\Theta_M\epsilon)\| \lesssim \frac{\pi \cdot (\sigma_\epsilon + 2E)\pi\sigma_w\sigma_\epsilon\sqrt{\frac{p\log p}{rn}}}{2(\sigma_w^2 + \sigma_w\sigma_x)\|\boldsymbol{\beta}\| + 2(\sigma_w^2 + \sigma_w\sigma_x + \sigma_x^2)E + 2(\sigma_w + \sigma_x)\sigma_\epsilon} \tag{22}$$

$$\|(\Theta_M\mathbf{X})^\top(\Theta_M\mathbf{W})\boldsymbol{\beta}\| \lesssim \frac{\pi \cdot (\sigma_\epsilon + 2E)\sigma_x\sigma_w\|\boldsymbol{\beta}\|\sqrt{\frac{p\log p}{rn}}}{2(\sigma_w^2 + \sigma_w\sigma_x)\|\boldsymbol{\beta}\| + 2(\sigma_w^2 + \sigma_w\sigma_x + \sigma_x^2)E + 2(\sigma_w + \sigma_x)\sigma_\epsilon} \tag{23}$$

$$\|(\Theta_M\mathbf{W})^\top(\Theta_M\mathbf{W})\boldsymbol{\beta}\| \lesssim \frac{\pi \cdot (\sigma_\epsilon + 2E)\sigma_w^2 \left(C\sqrt{\frac{p\log p}{rn}} + \sqrt{p}\right)\|\boldsymbol{\beta}\|}{2(\sigma_w^2 + \sigma_w\sigma_x)\|\boldsymbol{\beta}\| + 2(\sigma_w^2 + \sigma_w + 1)E + 2(\sigma_w + 1)\sigma_\epsilon}. \tag{24}$$

We observe that each of the quantities in Eqs. (22 - 24) are scaled by a term proportional to

$$\frac{\pi \cdot (\sigma_\epsilon\sigma_x + 2\sigma_x^2 E)}{2(\sigma_w^2 + \sigma_w\sigma_x)\|\boldsymbol{\beta}\| + 2(\sigma_w^2 + \sigma_w\sigma_x + \sigma_x^2)E + 2(\sigma_w + \sigma_x)\sigma_\epsilon}. \tag{25}$$

Taking the limit of large $E$ of the above (see remark 8) and setting $\sigma_x = 1$ we get

$$\pi^* = \lim_{E\to\infty} = \frac{\pi}{(\sigma_w^2 + \sigma_w)}.$$

Replacing the scaling factor in Eq. (25) with $\pi^*$ completes the proof. $\qquad\square$

**Remark 8** (Taking $\lim_{\|\widehat{\boldsymbol{\beta}}_{\text{OLS}}-\boldsymbol{\beta}\|\to\infty}$)**.** *Intuitively, when $E = \|\widehat{\boldsymbol{\beta}}_{OLS} - \boldsymbol{\beta}\|$ is small, this suggests that the effect of the corruptions is negligible and the full (or subsampled) least squares solution is close to optimal. Alternatively, when $E$ is large, the corruptions have a large effect on the estimate and so influence subsampling should work well. Note that here the size of $E$ is dependent on $\sigma_w$ and $\pi$. If we send $E \to \infty$ by allowing many points to be corrupted, the relative performance of IWS-LS compared with OLS worsens. However if we allow $\sigma_w$ to be large, the relative performance of our method improves.*

## SI.6 Supporting concentration inequalities

Here we collect results which are useful in the statements and proofs of our main theorems. Aside from Corrolary 12 which is a simple modification of Lemma 11, we defer the proofs to their original papers.

**Lemma 11** (Originally Lemma 25 from [4]). *Suppose* $\mathbf{X} \in \mathbb{R}^{n \times k}$ *and* $\mathbf{W} \in \mathbb{R}^{n \times m}$ *are zero-mean sub-Gaussian matrices with parameters* $(\frac{1}{n}\Sigma_x, \frac{1}{n}\sigma_x^2)$, $(\frac{1}{n}\Sigma_w, \frac{1}{n}\sigma_w^2)$ *respectively. Then for any fixed vectors* $\mathbf{v}_1$, $\mathbf{v}_2$, *we have*

$$\mathbb{P}\left[|\mathbf{v}_1^\top \left(\mathbf{W}^\top \mathbf{X} - \mathbb{E}[\mathbf{W}^\top \mathbf{X}]\right)\mathbf{v}_2| \geq t\|\mathbf{v}_1\|\|\mathbf{v}_2\|\right] \leq 3\exp\left(-cn\min\left\{\frac{t^2}{\sigma_x^2\sigma_w^2}, \frac{t}{\sigma_x\sigma_w}\right\}\right) \quad (26)$$

*in particular if* $n \gtrsim \log p$ *we have w.h.p.*

$$|\mathbf{v}_1^\top \left(\mathbf{W}^\top \mathbf{X} - \mathbb{E}[\mathbf{W}^\top \mathbf{X}]\right)\mathbf{v}_2| \leq \sigma_x\sigma_w\|\mathbf{v}_1\|\|\mathbf{v}_2\|\sqrt{\frac{\log p}{n}}$$

*Setting* $\mathbf{v}_1$ *to be the first standard basis vector, and using a union bound over* $j = 1, \ldots, p$, *we have w.h.p.*

$$\|\left(\mathbf{W}^\top \mathbf{X} - \mathbb{E}[\mathbf{W}^\top \mathbf{X}]\right)\mathbf{v}\|_\infty \leq \sigma_x\sigma_w\|\mathbf{v}\|\sqrt{\frac{\log p}{n}}$$

*holds with probability* $1 - c_1\exp(-c_2\log p)$ *where* $c_1, c_2$ *are positive constants which are independent of* $\sigma_x, \sigma_w, n$ *and* $p$.

**Corollary 12** (Modification of Lemma 11 for $n = 1$). *Suppose* $\mathbf{X} \in \mathbb{R}^{n \times k}$ *and* $\mathbf{W} \in \mathbb{R}^{n \times m}$ *are zero-mean sub-Gaussian matrices with parameters* $(\frac{1}{n}\Sigma_x, \frac{1}{n}\sigma_x^2)$, $(\frac{1}{n}\Sigma_w, \frac{1}{n}\sigma_w^2)$ *respectively. Then for any fixed vector* $\mathbf{v}_1$ *and* $n = 1$ *we have w.h.p.*

$$\|\left(\mathbf{W}^\top \mathbf{X} - \mathbb{E}[\mathbf{W}^\top \mathbf{X}]\right)\mathbf{v}\|_\infty \leq \sigma_x\sigma_w\|\mathbf{v}\|\log p.$$

*Proof.* Setting $t = c_0\sigma_x\sigma_w\log p$, $n = 1$ and $\mathbf{v}$ as the first standard basis vector in Inequality (26) in Lemma 11 and applying a union bound over $j = 1, \ldots, p$ yields the result. $\square$

**Lemma 13** (Originally Lemma 11 from [5]). *If* $\mathbf{X}$ *and* $\mathbf{W}$ *are zero-mean sub-Gaussian matrices then*

$$\mathbb{P}\left[\sup_{\|\mathbf{v}_1\|=\|\mathbf{v}_2\|=1}|\mathbf{v}_1^\top \left(\mathbf{W}^\top \mathbf{X} - \mathbb{E}[\mathbf{W}^\top \mathbf{X}]\right)\mathbf{v}_2| \geq t\right] \leq 2\exp\left(-cn\min(\frac{t^2}{\sigma_x^2\sigma_w^2}, \frac{t}{\sigma_x\sigma_w}) + 6(k+m)\right)$$

*In particular, for each* $\lambda > 0$, *if* $n \gtrsim \max\left\{\frac{\sigma_x^2\sigma_w^2}{\lambda^2}, 1\right\}(k+m)\log p$, *then w.h.p.*

$$\sup_{\mathbf{v}_1,\mathbf{v}_2} |\mathbf{v}_1^\top \left(\mathbf{W}^\top \mathbf{X} - \mathbb{E}[\mathbf{W}^\top \mathbf{X}]\right)\mathbf{v}_2| \leq \frac{1}{54}\lambda\|\mathbf{v}_1\|\|\mathbf{v}_2\|.$$

**Lemma 14** (Quadratic forms of sub-Gaussian random variables. Theorem 2.1 from [11]). *Let* $\mathbf{A} \in \mathbb{R}^{n \times n}$ *be a matrix, and let* $\boldsymbol{\Sigma} := \mathbf{A}^\top \mathbf{A}$. $\mathbf{x}$ *is a mean-zero random vector such that, for some* $\sigma \geq 0$,

$$\mathbb{E}\left[\exp(\alpha^\top \mathbf{x})\right] \leq \exp(\|\alpha\|^2\sigma^2/2)$$

*for all* $\alpha \in \mathbb{R}^n$. *For all* $t > 0$

$$\mathbb{P}\left[\|\mathbf{A}\mathbf{x}\|^2 > \sigma^2\left(\text{tr}\left(\boldsymbol{\Sigma}\right) + 2\sqrt{\text{tr}\left(\boldsymbol{\Sigma}^2\right)t} + 2\|\boldsymbol{\Sigma}\|t\right)\right] \leq e^{-t}.$$

**Lemma 15** (Extremal singular values of a matrix with i.i.d. sub-Gaussian rows. Theorem 5.39 of [18]). *Let* $\mathbf{A}$ *be an* $n \times p$ *matrix whose rows* $\mathbf{A}_i$ *are independent sub-Gaussian isotropic random vectors in* $\mathbb{R}^p$. *Then for every* $\tau \geq 0$, *with probability at least* $1 - 2\exp(-c\tau^2)$ *we have*

$$\sqrt{n} - C\sqrt{p} - \tau \leq \sigma_n(\mathbf{A}) \leq \sigma_1(\mathbf{A}) \leq \sqrt{n} + C\sqrt{p} + \tau$$

*where* $C$ *and* $c$ *are constants which depend only on the sub-Gaussian norm of the rows of* $\mathbf{A}$.

### SI.6.1  Discussion

In this section we provide some additional discussion about the bias and variance of our influence weighted subsampling estimator compared with known results from [5]. We first reproduce the following Lemma

**Lemma 16** (Originally Corollary 4 from [5]). *If $\Sigma_w$ is known and $n \gtrsim \frac{(1+\sigma_w^2)^2}{\lambda_{\min}(\Sigma_x) p \log p}$. Then w.h.p., plugging the estimator built using $\widehat{\Sigma} = \mathbf{Z}^\top \mathbf{Z} - \Sigma_w$ and $\hat{\gamma} = \mathbf{Z}^\top \mathbf{y}$ into Lemma 10, satisfies*

$$\|\widehat{\boldsymbol{\beta}} - \boldsymbol{\beta}\| \lesssim \frac{(\sigma_w^2 + \sigma_w)\|\boldsymbol{\beta}\| + \sigma_\epsilon \sqrt{1+\sigma_w^2}}{\lambda_{\min}(\Sigma_x)} \sqrt{\frac{p \log p}{n}} \; . \tag{27}$$

*When only an upper bound $\bar{\Sigma}_w \succeq \Sigma_w$ is known then*

$$\|\widehat{\boldsymbol{\beta}} - \boldsymbol{\beta}\| \lesssim \frac{\left[(\sigma_w^2 + \sigma_w)\|\boldsymbol{\beta}\| + \sigma_\epsilon \sqrt{1+\sigma_w^2}\right]}{\lambda_{\min}(\Sigma_x) - \lambda_{\max}(\bar{\Sigma}_w - \Sigma_w)} \sqrt{\frac{p \log p}{n}} + \frac{\lambda_{\max}(\bar{\Sigma}_w - \Sigma_w)\|\boldsymbol{\beta}\|}{\lambda_{\min}(\Sigma_x) - \lambda_{\max}(\bar{\Sigma}_w - \Sigma_w)} \; . \tag{28}$$

We can compare these two statements with our result from Theorem 1. Eq. (27) is similar to the bound we have from Theorem 1 up to the bias term assuming $\pi = 1$ (i.e. all of the points are corrupted). Since we do not use knowledge of $\Sigma_w$ we can compare our result with Eq. (28) which has a bias term which is related to the uncertainty in the estimate of $\Sigma_w$ which in our case is $\sigma_w^2$. It is clear from Lemma 16 that the only way to remove this bias completely is to use additional information about the covariance of the corruptions.

## SI.7    Additional results

In this section we provide additional empirical results.

**Non-corrupted data.**    We first compare performance in three different leverage regimes taken from [13]: uniform leverage scores (multivariate Gaussian), slightly non-uniform (multivariate-t with 3 degrees of freedom, T-3), highly non-uniform (multivariate-t with 1 degree of freedom, T-1). Full details of the data simulating process can be found in [13].

Figures 3 and 4 show the estimation error and the RMSE respectively for the simulated datasets described in [13]. The results for the T-3 data are similar to the Gaussian data. The slightly heavier tails of the multivariate $t$ distribution with 3 degrees of freedom cause the leverage scores to be less uniform which degrades the performance of uniform subsampling relative to `SRHT-LS` and `IWS-LS`. Figure 4 shows that the RMSE performance is similar to that of the statistical estimation error.

(a) Gaussian

(b) T-3

(c) T-1

Figure 3: Comparison of mean estimation error and standard deviation on a selection of non-corrupted datasets.

**Corrupted data.**    Figures 5 and 6 show the estimation error and RMSE respectively for the corrupted simulated datasets. In all settings influence based methods outperform all other approximation methods. For $5\%$ corruptions for a small number of samples `ULURU` outperforms the other subsampling methods. However, as the number of samples increases, influence based methods start to outperform OLS. For $> 3000$ subsamples, the bias correction step of `ULURU` causes it to diverge from OLS and ultimately perform worse than uniform.

For $10\%$ corruptions, `aIWS-LS` and `aRWS-LS` converge quickly to `IWS-LS`. As the number of corruptions increase further, the relative performance of `IWS-LS` with respect to OLS decreases slightly as suggested by Remark 8.

(a) Gaussian

(b) T-3

(c) T-1

Figure 4: Comparison of root mean squared prediction error (RMSE) and standard deviation on a selection of non-corrupted datasets.

For 30% corruptions, the approximate influence algorithms achieve almost exactly the same performance as `IWS-LS`. Even for a small number of samples all of the influence methods far outperform OLS. As the proportion of corruptions increases further, the rate at which the approximate influence algorithms approach `IWS-LS` slows and the relative difference between `IWS-LS` and OLS decreases slightly. In all cases, influence based methods achieve lower-variance estimates. Here, `ULURU` converges quickly to the OLS solution but is not able to overcome the bias introduced by the corrupted datapoints.

(a) 5% Corruptions

(b) 10% Corruptions

(c) 30% Corruptions

Figure 5: Comparison of mean estimation error and standard deviation on a selection of corrupted datasets.

(a) 5% Corruptions

(b) 10% Corruptions

(c) 30% Corruptions

Figure 6: Comparison of test RMSE and standard deviation on a selection of corrupted datasets.

**Larger Scale Experiments Corrupted data.** We now present results on larger scale simulated data. We used the same experimental setup as in §7 but we increase the size of the data to $n = 100,000$ and $p = 500$.

Figures 7 and 8 show the estimation error and RMSE respectively. In this setting, computing IWS-LS is too slow (due to the exact leverage computation) so we omit the results but we notice that aIWS-LS and aRWS-LS quickly improve over the full least squares solution and the other randomized approximations. The general trend in this setting is the same as with the smaller experiments, however for $5\%$ corruptions the improvement of aIWS-LS and aRWS-LS over OLS happens with a much smaller subsampling ratio than with smaller datasets.

(a) 5% Corruptions

(b) 10% Corruptions

(c) 30% Corruptions

Figure 7: Comparison of mean estimation error and standard deviation on a selection of corrupted datasets.

(a) 5% Corruptions

(b) 10% Corruptions

(c) 30% Corruptions

Figure 8: Comparison of test RMSE and standard deviation on a selection of corrupted datasets.