[Reviews · NeurIPS 2014]

Submitted by Assigned_Reviewer_8

This paper looks at alternatives to LS for solving linear regression
problems. Classical statistics will consider a score function as a
way of weighting observations. But, these would only capture an
additive error which is different than Gaussian. This paper instead,
also considers the influence of a point. Statisticians for years have
thrown out influential outliers--but as far as I can find, this is the
first automatic procedure that uses influence as a way of
down-weighting a regression.

The probabilistic model presented here is not the driving force of the
paper. In other words, it is pretty easy to identify which points are
contaminated and which are clean. (This is very easy for large p
since the two distributions won't overlap.) So if the estimator
presented was tied to this model, I would be unexcited about the
results. But, the estimator is fairly natural. So the model is only
being used as an example.

1) Put a line where the accuracy of the MLE for your probabilistic
model would be. You shouldn't end up being close to this line since
you aren't assuming your model. But it still is a more useful
comparison point than the LS line. If you don't want to bother with
doing the de-convolution problem, you can assume that the ML has
access to the random variables U. Then it will be a lower bound on
how well it will perform. As p --> infinity, this lower bound will
become tight.

2) Make the connection to some of the literature on the use of scoring
functions (Since this goes back to the early 1970's I'm not going to
suggest any literature). For example, the aRWS can be thought of as
down weighting points by a 1/epsilon^2. If one were to down weight by
1/epsilon, that would correspond to doing a L-1 regression. So it
would be similar to assuming a double exponential distribution for the
errors. Your weighting function is even more draconian. So what
distribution does it correspond to if you think of it as a score?
(I'm guessing it looks like a Cauchy distribution--but only
approximately.) (Note: you are sampling, whereas I'm more used to
thinking of weights. So it might be that your 1/epsilon^2 sampling is
related to a 1/epsilon weight. If so, then it is approximating a L-1
regression. This would be very nice to point out if it were true.)

3) Are you claiming to be the first people to suggest weighting by
1/influence? This might be true--but it is a strong claim.

4) I think what you are doing is using a different regression function
for "large observations" than you use for "small observations." One
way to test this and provide a better comparison for LS would be to
define tilde-l as you do and then interact this variable with all your
X's in your regression. This will allow the LS methods to have access
to tilde-l. Hence LS should be able to end up with a better estimator
since it now can have different slopes for the pure X observations
than it has for the X+W observations.

5) There is extensive literature on both errors in variables and
robust regressions. At least a few of these 1000+ papers should have
something useful to say. Put in at least some effort into finding
something that connects. That will help the implied claim that your
methods are in fact new.

6) PLease look at this 1980's NBER paper:

http://www.nber.org/chapters/c11698.pdf

In particular, equations (15) and (16). This is very close to the estimator you are using. It would be nice if you were to chase down some of the modern references and see if there are any connections to your work.

Summary: This paper provides a new alternative to robust regression. Namely, it down weights by the influence of a point. A useful theorem is provided and good alternatives are considered in the empirical section.

Submitted by Assigned_Reviewer_23

- Summary:
This paper presents a speed-up proposal for least squares linear regression. Compared to existing approaches the current proposal claims robustness to outliers or corruptions in the measured predictors.

- Quality:
The paper seems technically sound to me. The claims are well supported from a theoretical perspective. From an experimental perspective I was a bit disappointed. The initial motivation of the authors was estimation on large scale settings. Although in terms of artificial data sets the authors have used reasonably large data (n=100,000; p=500), regards the single real world domain it can hardly be considered a large data set by today standards (n=13,000; p=170). Moreover, as mentioned before, there was a single real world data set. This is a bit limiting in terms of experimental results, particularly given the initial motivation immediately outlined in the first sentence of the abstract. Still, the work is very complete from a theoretical perspective, particularly when we take into account the supplemental material that was provided.

- Clarity:
I think the paper reads very well. The algorithms and methodological approach are clear and I think it should be easy to reproduce the results of the authors. Still, a bit more details on the Airline delay dataset would be preferable (still an URL is given...). I also do not understand the label on the X-axis of figure 2 (n_{subs})... what is the meaning of this label? On page 5, near Eq. 7, there seems to be some sort of typo (... $4 and $4 ...)

- Originality:
The proposal seems novel to me. Still, the proposal is a combination of previous works ([8] and [14] as the authors refer on section 5) and thus with limited novelty. Nevertheless, the authors are clear about this and identify relevant and related works.

- Significance:
The results reported in the paper are immediately limited by the fact that they are constrained to a specific type of regression method. Moreover, given the limitations on the experimental evaluation mentioned above, I'm afraid it is also not easy to completely assert the impact of these proposals in terms of experimental results. This means that although I think this is an advance on the state of the art of least squares linear regression, the exact impact of this advance is not correctly evaluated by the authors in my opinion, which limits the significance of the paper that was already potentially small by being constrained to a specific regression algorithm.

Summary: This paper presents a very localized contribution that seems theoretically sound but whose impact is limited both because of its focus on a single technique and by a limited experimental evaluation

Submitted by Assigned_Reviewer_25

Summary:

The paper presents an influence reweighted sampling method (IWS-LS) (as well as a residual weighted sampling method, i.e., RWS-LS) for learning large-scale least squares, which is robust with respect to some data corruption, e.g., the particular sub-Gaussian additive noise. Existing approximation methods are adopted to compute the OLS estimate and the leverage scores. Estimation error is analyzed theoretically for IWS-LS. Finally, empirical results are reported on both synthetic and real-world data sets.

Comments:
Overall, the paper is well written. The influence reweighted subsampling method is new. The theoretical and empirical results appear to be sound.

For the experiments, the dataset with 100,000 samples in a p=500 space is not huge (the real-world Airline delay dataset is even smaller). The computation of the OLS estimate as well as the leverage scores should be feasible (with O(n p^2) complexity), even though it may take minutes or hours, on a standard desktop; and thus the results of exact methods should be included as a baseline. It is also helpful to include the experiment environment. Furthermore, as time efficiency is one very important aspect of large-scale learning, running time should be included.

For data nosing/corruption, this paper focuses on developing estimators that are robust to the resulting outliers. It would be useful if the authors can discuss on the work that leverages data nosing to actually improve classification/regression. Some recent work includes dropout training for deep neural networks [1,2], learning with marginalized corrupted features [3,4], and etc.

References:
[1] G. Hinton et al., Improving neural networks by preventing co-adaptation of feature detectors. arXiv:1207.0580v1, preprint.
[2] S. Wager, et al., Dropout training as adaptive regularization. NIPS, 2014.
[3] L. van der Maaten, et al., Learning with marginalized corrupted features. ICML, 2013.
[4] N. Chen, et al. Dropout Training for Support Vector Machines, AAAI, 2014.

Finally, some typos should be corrected. For example, line 42 “more more realistic”; line 254 “ideas from $4 and $4”.
Summary: The paper is well written. The influence reweighted subsampling method is new. The theoretical and empirical results appear to be sound.
Author Feedback
Author rebuttal: We thank the reviewers for their helpful comments. We address
the major criticisms and suggestions in turn.

“The results ... are constrained to a specific type of regression method.”
We argue our contribution has much wider implications since
1. Linear regression is ubiquitous. We present scalable
approximation schemes that are robust against quite a
general form of corruptions. aRWS and aIWS can also easily
be applied to logistic-regression.
2. In Thm. 9 we show that for uncorrupted data our proposed estimator
achieves the same rate of convergence as Uluru suggesting that IWS
is applicable in most situations where randomized approximations
are. We also verify this empirically. For heavy tailed data (where
our assumptions are violated), we show that IWS does not perform as
well as Uluru, nevertheless aRWS still performs well.
3. We have developed some new theoretical insights into classical
regression diagnostics.

Size of datasets: The absolute size of a data set should not change
the behaviour and comparison of the algorithms. Important is the ratio
of samples to features. The trend will be similar for fixed
ratio. Figures in the supp show similar results when n=10,000 which
suggests similar scaling for different n. Also, our experiments
represent an order-of-magnitude increase in size over those reported
in previous related works i.e. [5, 7, 14]. This combined with the
released software demonstrate big steps towards scalability of our
algorithm.

“label on the X-axis of figure 2” n_{subs} is the number of observations subsampled to be used in the regression. It is also the number of observations used in
the SRHT to approximate the influence, leverage scores and residuals. We report the performance of our estimator compared to other subsampling estimators as a function of the number of samples used, and full least squares.

“The results of exact methods should be included as a baseline...
running time should be included.“
We have included the least squares fit line (“full” in Fig 2.). IWS is included in 2(c) and in Figs 3-6 in the supplement. We will add the experimental settings. Fair benchmarking is difficult: low-level linear algebra routines for solving least squares are highly optimized and multi-threaded whereas our software is is not.
Therefore, additional overhead when applying random projections make absolute timings difficult to interpret. Some works (eg [7]) skirt this issue by plotting error against “theoretical FLOPS” which we avoid. A link to full source code for our algorithms and the comparisons are provided for reproducibility.

“discuss the work that leverages data noising”
The reviewer raises an interesting point. In fact, independently
we are actively investigating the connection between the dropout
regularizer of Wager et al., and influence: both reweight observations
based on their usefulness for prediction. Perhaps these ideas can be
used to extend our results towards a more general idea of robust estimators.

“it is pretty easy to identify which points are contaminated”
The observation that the proposed algorithms are not tied
to this model is correct; they will also likely improve over
leverage-sampling methods for non-Gaussian noise distributions for
instance. We argue that even in high dimensions, it is difficult to
tell the corrupted points from the non-corrupted points using leverage
alone (i.e. the difference between lemma 5 & 7), as illustrated
in Fig 1.

“Put a line where the accuracy of the MLE for your probabilistic model
would be…”
We thank the reviewer for this good suggestion. We’ll add the
suggested lower bound. To clarify our choice of comparisons: we wanted
to compare against only methods we can estimate from data alone. We
could estimate the MLE under the corrupted model using e.g. [12] but
since we do not a-priori know the covariance of the corruptions, this
is not a feasible comparison.

“it might be that your 1/epsilon^2 sampling…is approximating a L-1
regression. This would be very nice to point out if it were true.”
Very interesting! Given the time constraints we are unable to verify
this theoretically, but your observation about aRWS is deserving of a
proper analysis which we defer to future work. As a first thought:
empirical results in figure 3(c) show that for heavy tailed data, aRWS
performs well, which supports the reviewers suggestion that it is
close to a robust, L-1 regression.

“Are you claiming to be the first people to suggest weighting by
1/influence?”
To our knowledge we develop the first finite sample bounds for
influence and are the first to use it as an importance sampling
distribution in this context. The literature on randomized least
squares is relatively young. A recent arXiv paper references
“Influence sampling for generalized linear models” by Jia et al. but
is not yet available and we haven't seen it.

“a better comparison for LS...”
This seems similar in spirit to the approach of [12] etc except where
one would estimate the covariance of the corruptions from data? This
is an interesting idea and worth pursuing.

Related literature & 1980's NBER paper
Thanks. Our setting is robust estimation in scenarios where computing
the full least squares solution is impractical. This differs from the
classical literature which often requires the computation of multiple
full least squares solutions. Eq (15, 16) in the NBER paper are
related in that they downweight influential points whereas we sample
these points with low probability. The important difference is that
1) we never compute the full influence (but approximate it) and 2)
through sub-sampling we solve a smaller, computationally feasible
problem. We thank the reviewer for helping us connect to the related
classical literature. It may be possible to introduce similar
approximation schemes for many of the classical robust M-estimators.